# Church Governance—A Philosophical Approach to a Theological Challenge in an Anglican Context

Peter D. G. Richards 

Independent Researcher, Melbourne, VIC 3163, Australia; ethics.peter@gmail.com

**Abstract:** Church governance is not often debated within a philosophical or theological sphere. This is perhaps because church governance has been part of tradition since Constantine and the initial Greek philosophical world view of sovereignty and hierarchy. Such a stance has led towards a managerial mindset that follows and conforms to the world, which plays out within the Anglican polity in the setting of an adversarial parliamentary style synod. This style encourages bounded communities of power that often refute the burgeoning inspirations of the Spirit. In changing the underlying theological basis of such a stance, by invoking the understanding of an undeniable community in the singularity of the Triune God, governance becomes more open. Engaging with, primarily, Agamben but also others from philosophy, a new viewpoint is presented to challenge the manner through which tradition is wielded as the only possibility. In seeing through a differing lens, communities can be conceived as both porous and interconnected, thus allowing the body of Christ to respond with transformative action as opposed to a continuum of conformance with secular legality. In this manner, the bishop's role may become more centralised towards a Eucharistic one, as opposed to the managerial mindset and role, to enhance the possibilities of God's love. This then removes the need for a hierarchy driven by a sovereign mindset that tradition bolsters, whilst maintaining loving and authoritative oversight that tradition suggests.

**Keywords:** governance; church; Anglican; philosophy; theology

## 1. Introduction

It would perhaps be appropriate to ask what place church governance has in a Special Edition on theology and European philosophers, especially when considered in terms of a specific denominational context. This is particularly true when we think about governance as being a secular matter and not something of great theological concern or reflection, as it appears not to touch on matters of faith, dogma, etc. This is not to say that religious faith, spirituality, and the effect of religion on management and governance are not researched. Indeed, much has been written with regard to religion and business (Alewell et al. 2023), yet most papers do not touch on the underlying theology and its interaction with management processes. For example, accountability, a cornerstone of governance, has been examined from the spiritual dimension (Alewell et al. 2023, p. 102) as well as from the faith dimension (Keplinger and Feldbauer-Durstmüller 2023), rather than from a theological context. Theological input is seen when business practices are scrutinised from an ethical viewpoint, with reference to religious ethical teachings and its effect on a corporation (Cremers 2017), but looks more at human anthropology rather than deeper theology in terms of governance per se. The most recent theological input into the governance debate has come from Deslandes, who has looked at Gianni Vattimo and John Caputo's work on "weakness" and theological *oikonomia* in the practice of secular management (Deslandes 2018), as opposed to the governance of the church.

Furthermore, it would appear to have little current connection to philosophy, except, perhaps, through the question of sovereignty entangled with power/politics. Whilst the tangent of sovereignty is somewhat oblique, it has a major impact upon how the church has

been and is being governed. Political entanglement with power and sovereignty appears early in the structures of the church, leading towards a certain amount of epistemic hubris, both in the Anglican polity (into which polity I fall) and Catholicism, which manifests in a leadership who believes that they have "*the* answer to the problem or that only they can solve it" (Ogden 2018, p. 53). Such hubris arises from a specific construction of an episcopal calling from 'God alone' that is not open to inquiry from within or outside the structures of the church. The basis of this governance framework is founded, in part, upon an understanding of a monotheistic, all powerful paternal deity; the spiritual works of Dionysus the Aeropagite; and the Constantinian disposition of power, as the newly embraced church became part of the empire. The construct of church governance that Derrida suggests is an "un-avowed theologism" (Derrida 2005, p. 110) and is an implicit paradigm which advocates and fosters an understanding of monarchical power and sovereignty over the realm of organisational governance. In the Anglican polity, this creates a disputed space that is allowed to operate through confrontational politics embedded within the synod, rather than creating a means to advocate for and be guarantors of God's love in the world.

The word governance is derived from the etymological root of the word κυβερνάω (*kubernáo*), which means "to steer or pilot a ship/chariot" (European Commission 2002), with perhaps an understanding that such steerage and pilotage is towards the guarantee of a safe passage over stormy seas. This meaning has changed and morphed within the world, moving away from the original intent of a steersman using a rudder to guide the ship through the sea. It has now become the embodiment of the slave-driving captain enforcing his will through the whip, which governance is now perceived to be. In becoming the whip, governance requirements change at a fast-moving pace to keep up with the general expectations of the secular public (the owner), whilst becoming increasingly entangled with power, politics, authority, and community structures. To regain control of its governance and return control back to an ever-loving and forgiving God, consideration needs to be given to how the world has changed, whilst interacting with it in a manner that delivers God's promises of justice, peace, and love.

The intention here is to engage the imagination, achieving a moment of hopeful inspiration, awakening "*gelassenheit*" as perhaps Heidegger might call it. A moment where we allow ourselves to be released from our captive and historically common stance into a new place of understanding for those that are in leadership positions within the structures of the church. By engaging with the thoughts of several European philosophers, this paper seeks to disrupt the historical paradigm by involving alternative understandings that would increase the role of a theological impost on governance in the church to steer it back into deeper and calmer waters, whilst embodying a Christlikeness within the structures that allow governance to happen. All denominations have arisen as a result of the manoeuvring of political power and authority to attain the desired outcomes of their founders. The worldwide Anglican Communion is as influenced as any other by its tradition and has inherited a structure of governance which includes the threefold ministry of bishop, priest, and deacon. Being part of the governance structures of the Australian Anglican Church, much of the discussion and thoughts are within the context of the broader Anglican Communion as a denomination but are applicable to any other, with the understanding that office holders may have differing titles but often wield similar power.

## 2. Common Characteristics of Governance

It is also of importance to note, at this point, that, in governance, there is thought to be "seven characteristics of good corporate governance", which could be seen to apply to all governance structures, which have been most succinctly defined by the King Report from South Africa (King Committee on Corporate Governance 2002). These seven characteristics are (i) discipline, (ii) transparency, (iii) independence, (iv) accountability, (v) responsibility, (vi) fairness, and (vii) social responsibility (King Committee on Corporate Governance 2002, pp. 10–11). The South African report by Justice King was probably one of the first

governance reports that emphasised these characteristics, following the Cadbury report in 1992; it also went further by suggesting that these were not just the board's responsibility, but extended to the whole company in its approach to governance (King Committee on Corporate Governance 2002). Whilst other jurisdictions have followed similar lines, they have not always directed governance towards the whole organisation in the manner of the King report and its subsequent updates, which have included the concept of operating with *Ubuntu* (King Committee on Corporate Governance 2016); with regards to the former, see, for example, the Australian Securities Exchange document on good governance (ASX Corporate Governance Council 2019). These eight principles are seen as the standards to which corporate office bearers are held accountable to the shareholders and public. Within the Christian sphere, these standards can be equated to a practical implementation of the beatitudes in church governance. They are principles that should underpin church governance irrespective of jurisdiction or denomination and be theologically incorporated into such governance.

Discipline is seen as being the adherence to behaviours that are accepted as being right and ethical on a universal basis. This would mean a commitment to governance itself and the ability to self-manage the organisation's board in terms of the role and responsibility of each member. Transparency is the ease with which those outside the organisation can discern a true understanding of the organisation. This is understood to be undertaken by the organisation's release and timely disclosure of information by the board and the organisation to those that require such information (investors, legal bodies, etc.). Independence is typically shown by having mechanisms in place to ensure that appropriate appointments are made in a manner that avoids conflicts of interest, both internally at the board level as well as externally through the appointment of auditors, etc. Accountability inculcates the responsibility of each decision maker within the company to have effective processes in place that show why and how decisions have been made. This ensures that interested parties have a means to make inquiries and assess the decisions made. Responsibility means that there are clear behavioural codes and risk frameworks in place so that breaches can be managed to ensure that the corporation is maintaining its public and shareholder responsibilities. Fairness is the ability of the corporation to treat all its personnel and shareholders with equity for their rights (financial, social, and private) within the context of the corporation. This entails listening to the minority as much as to the majority, in terms of shareholders and other stakeholders. Social responsibility places a responsibility upon the corporation to respond to social, environmental, and human rights issues, with a high priority on its own ethical standards, with a consequential increase in indirect benefits.

Whilst some of these can clearly be seen to be emphasised in the church's governance, i.e., social responsibility, it is equally obvious that many of these characteristics are often lacking in ecclesial governance, as, perhaps, recent scandals have demonstrated.

## 3. A Common Progenitor—The Status Quo of Power and Authority in the Anglican Church

The Anglican Communion is a collective of independent provinces containing independent dioceses, each of which are governed in terms of spiritual and secular direction, through the processes of each synod and the ecclesial authority of each diocesan Bishop. Power and authority are, thus, held in a hierarchical structure, at the head of which is the Bishop of the diocese. This invests extraordinary secular power in the hands of the Bishop, which is easily embodied within the existing hierarchical power structures of the organisation. What is the derivation of such power (philosophically) and, perhaps more importantly, how does the background theological tradition influence the denomination's thinking of governance processes?

In his study of the genealogy of power in the West, Agamben notes a "double structure" to government that he initially segments with regard to *auctoritas* (authority) and *potestas* (power) in his book '*The State of Exception*', which is followed by an "articulation between

the Kingdom and Government" with specific regard to the relationship between "*oikonomia and Glory*" (*oikonomia*, i.e., economy) (Agamben 2005a, 2011). One of the issues he draws out is the different originators of political life in the modern era with particular regard to the foundations of modern political philosophy and biopolitics. The first of these arising out of the paradigm of political theology and the latter arising from economic theology, which replaces the sovereignty of God with "the idea of an *oikonomia*, conceived as an immanent ordering" which is, in turn, based upon the household, rather than politics (Agamben 2011, p. 1). The episcopally led church, in the main respect, has followed the former paradigm in terms of its internal governance, rather than the latter, which can be found displayed in the first formations of governance in the Pauline churches with their more egalitarian approach.

The burgeoning churches began governance with a clear egalitarian understanding of governance, as can be derived from Paul in Galatians (5:13–15) (amongst others). In interpreting this passage with regard to the Roman empire as the love that Paul proclaims is a "continual mindfulness in discerning, disobeying and unfreezing the antithetical *nomos* of self versus other" (Kahl 2014, p. 269). It becomes the basis of a re-organisation of community that is not reliant on self-boasting, through the system of "euergetism/benefactions" becoming instead a community that is "nonhierarchical, nonantagonistic, nonexclusive" with an increase in "horizontal mutuality and solidarity", one that does not consume, fight or compete against the other (Kahl 2014, pp. 269–70). This changes when it joins to empire at the time of Constantine, as its clergy became increasingly involved in "protecting the integrity and welfare of the empire" (Hovorun 2017, p. 158). This became pronounced after the Council at Nicaea with the adoption of the metropolitan model tailored to the administrative units of the Empire (Hovorun 2017, p. 60), thus embedding an understanding of hierarchy as part of the church and its systems of governance by its integration into the Constantinian empire's traditional bureaucratic structures. This is further compounded through the acceptance of the mystagogical structure laid down in the works of Dionysus the Aeropagite (Dionysius the Aeropagite 1897). Agamben suggests that this 'sacred power', i.e., hierarchy, is an evolution of the concept of *diakosmēsis* found in the Neoplatonic work of Proclus that is "to govern by ordering (or to order by governing)" (Agamben 2011, p. 154). The church accepted this ordering in a manner that is sacrosanct and indelibly imprinted onto its governance matrix through the threefold order of bishop, priest, and deacon. Political changes during the reformation period have moved some denominations away from such a structure, but it has been retained by most episcopally led denominations, including the Anglican Communion.

By accepting this paradigm, the episcopally led church has accepted a power, as Foucault understands it, that leads to oppression or domination if abused or overextended in its pursuit and operation. Over time, the physical presence of the sovereign becomes unneeded, being supplanted by "a tightly knit grid of physical coercions", that improves its efficacy over time, whilst, at the same time, increasing the numbers that are under its subjugation (Foucault and Gordon 1980, p. 106). In this manner, the laity, both historically and genealogically, have a long generational history that embeds the (mis)understanding of the divine institution of the church and the appointment of its leaders from God, thus granting those in ordained authority a semblance of sovereign power. The bishops, in their epistemic hubris, have, at times, utilised such power, irrespective of consent or circumstance, for ecclesial governance to their own political ends, thus exercising their "apostolic" tradition rather than their "conciliar" tradition, which is ignored or relegated (Ogden 2018, p. 54). Such a paradigm of ecclesial governance is suggested to come from the foundations based on "the transcendence of sovereign power on the single God" (Agamben 2011, p. 1), thus unifying authority in the "epistemological centrality of the bishops" (Ogden 2018, p. 56). This, in turn, leads to a belief, by those in authority, of a permission to make the final decision due to their determination as to what is true, irrespective of context, knowledge, and/or consent (Ogden 2018, p. 56).

This power distribution is, perhaps, derived from a world view based on the understanding of male dominance and patriarchy, rather than equality. In taking this political route, the church moved away from the initial predominance of *ecclesia* (assembly of the people) towards a domination of *kyriachy* (a social system(s) built around domination, oppression, and submission), from "*kyrakon*, i.e., belonging to the lord/master/father" (the etymological origin of 'Church') (Schüssler Fiorenza 2007, p. 78). In doing so, it has embedded within its spiritual hierarchy a system which naturally discriminates against the other, especially with regard to race, gender, and sexuality. The imperial understanding of self that this reflects "projects evil onto the 'others' who do not follow Christ, the poor, prostitutes, homosexuals, the feminists", etc. (Schüssler Fiorenza 2007, p. 143). The basis of Liberal democratic thought espouses the formulation of "the state and society based on sets of principles: equality, autonomy, liberty, toleration, neutrality" which contain the "moral *ought*" (people *ought* to be equal, autonomous, etc.) (De Roover 2012, p. 142). Whilst ideal, these principles do not often become reality as "profits always count more than people" (Crockett 2013, p. 104), which leads to the enactment of hierarchical/power- based policies that "benefit some sectors of the community and disadvantage others", for example a diocese or parish, for the sake of power and/or authoritative position (Brown 2020, p. 24). The more selfish route elevates one above the other, as opposed to the Christian ethos of the other over the one.

*A Theological Underpinning of Hierarchical Structure*

Theologically, this could be said to be based heavily on an assumption of singularity at the start of the Genesis narrative. The initial phrase *Bereshit bara Elohim* (In the beginning, God created heaven and earth (Gen 1:1)) begins the narrative. This narrative "tells of an ordering and a goodness that shapes all the categories of creation" (Zornberg 2011, p. 3). In our interpretations, there is an assumption of a singularity at the beginning "Elohim" who then, alone, separates and hierarchically orders the "primordial mass into a 'good' pattern", thus bringing governance, through hierarchy, to chaos (Zornberg 2011, p. 3). This is the empiric ordering of the sovereign upon the disorder of chaos and each level of order is imposed from the Godhead in a hierarchical manner, as each conforms to the 'good' as understood by the unitary sovereign (Genesis 1:10b, 12b, 18b, 21b, 25b, 31). Here, we can see the necessary underpinnings of the *creatio ex nihilio* (creation out of nothing) doctrine, not really present in the phrasing of the first two verses, which emphasises the patriarchal sovereign leadership of society at the time and the dependence on this hierarchy in the ordering of society/governance. In this reading, we can see an understanding that could perhaps theologically underpin the current view of governance within the Anglican Communion and other denominations; that is, one based on a hierarchy of power and an imposition on those below what is viewed as being the best or what is good for the company/organisation/diocese. This, then, would suggest that the ordering of structure is undertaken from the upper echelons with little or no input from those below, as the one God is responsible for creation and that is seen to be good, especially as humankind has been handed responsibility to act for and on behalf of God (Gen 1:28). In this conception, the creativity that is found in the deep (*tehom*), which is *tohu vabohu* (formless chaos or void), is either classed as evil or as somewhere that needs to be put under control, so as not to escape (Keller 2003, p. 91). Such control is undertaken by a dominant governing patriarchal figure at the top that does not wish for the dangerous potential of creativity to intrude upon the order that has been imposed (Keller 2003, pp. 90–91). Thus, from a governance understanding, the bishop/archbishop/Pope are considered to be infallible in terms of how the organisation is run. In this manner, the status quo is managed and innovation is not encouraged unless it is considered 'good' by the hierarchy. In taking this route, there is an apparent discrepancy in terms of the key characteristics of good governance, especially when transparency and independence are examined.

## 4. An Uncommon Inheritance—Moving into New Life

Is it possible to re-conceive the route that governance has taken by looking at our interpretive effort, in conjunction with what a few selected modern philosophers tell us about the community, politics, and power. What happens when there is a reversal of the dominance of the traditions such that conciliarity is promoted over apostolicity? Modern corporations are more prone to a diverse governance structure, as are modern communities, who rely more and more on the preponderance of a culture of networked solutions to governance (Sørensen and Torfing 2003). In Anglican denominational governance, communities are often designated, as such, by being 'bounded communities', those being communities within a specific territory marked by a physical or hard ideological boundary often marked and labelled, i.e., parish or diocese; Anglican/Catholic/Muslim. It is precisely this bounded community that led Derrida to dislike the term community, which derives from a military-style fortification, and is precisely what he is aiming at, as regards deconstruction, with the "affirmation—*viens, oui, oui*—of the *tout autre*" (Caputo 1996, pp. 25–26). In allowing porosity within the bounds of the tight fortress of the bounded community, new ideas and new formulations are accepted and developed.

Whilst the bounded community still exists, especially within the Anglican Communion, today's community tends to be more porous, moving towards and encompassing the understanding of the communal nature of humanity, that "[T]o be in relation is already to be a multiplicity" (Keller 2011, p. 81). The understanding of multiplicity is that the individual as a singularity cannot exist, as one cannot exist, without a relationship (Schneider 2008, p. 143). This is underscored when it is realised that, mathematically, the integer 1 (one) is "fully dependent on its relation to and distinction from all other numbers" and, as such, "one comes into being in relation to Other/s, or not-ones" (Schneider 2008, p. 143). This is perhaps a useful reminder to us that the Christian faith worships one God, who is a Trinity, and, thus, in relation, for all oneness is but a relational reality. The porosity of the modern community, with its many modes of connection, is reminiscent of the imagery in the thought of Deleuze and Guattari as they speak of the "rhizome" (Deleuze and Guattari 1987, p. 6). In viewing a community as rhizomatic, the suggestion is of an ever-expanding network of inter-related nodes across space and time. This is very reminiscent of the interconnected web of the online community, which has been emphasized following the recent COVID-19 pandemic. The description of a community in this rhizomatic manner creates a community that is truly unbound, as the rhizome "ceaselessly establishes connections" (Deleuze and Guattari 1987, p. 7) much like the modern social media network and some governance structures in corporations, blending and melding input from a multitude of sources. This not only ensures compliance, but generates new approaches or ideas for the propagation of the organisation's premises.

### A Theological Underpinning to Networked Authority

Re-looking at the same beginning passage of the Genesis tale discussed above, some traditional Judaic interpreters, such as Rashi, suggest that there is no suggestion of sequence and that the "opening sentence tells us nothing about beginnings", as the true sentiment of *Bereshit* is not 'In the beginning" but rather "when" or alternatively "At" (Zornberg 2011, pp. 3–4; Keller 2003, p. 9). If such is the case, then the idea of *creatio ex nihilo* by a sovereign God of power at the head of a hierarchy is disrupted, as is a governance based on such sovereignty. The nothingness is not nothingness per se but *tohu vabohu*, which is dependent or associated with the deep (*tehom*) of, perhaps, uncertainty that has the potentiality of something new and abiding, over which the *ruach* (breath) of God hovers. This is a relational understanding that attracts newness and form which, in itself, leads from chaos into order, but has a dependency upon its initial conditions (Delio 2011, p. 26). Out of the "dialogic address" of God's breath, that is love, comes forth new life and new form from within the "womb of the *tehom*" (Kosman 2018, p. 10). This a more interactive and relational understanding of creation, moving away from the display of 'power over' (i.e., authoritative) that is present within a sovereign power, towards a 'power with' imagery.

In the use of the imagery of 'power with' (i.e., inclusive) and the "delinerarization of the time of creation", Jewish interpretation makes the chaos stuff of the *tohu vabohu* "neither nothing nor evil", but rather allows the solicitation of the formlessness into virtual form (Keller 2003, p. 115).

In this manner, instead of supressing innovation and the emergence of ideas within an organisation, there is a wooing of those who are able to connect with the inspiration to create newness of life for the organisation. In such a scenario, there is still a hierarchical platform, since there is no challenge to the presence of the unitary God, rather the challenge that is put forward is to the method by which the hierarchy manages its power and authority. This does not negate the understanding that the top of the hierarchy, the unitary sovereign, has the final determination as to what is good. It does mean that there is more availability for uniqueness to become 'good' in the eyes of the hierarchy as it is listened to and encouraged. There is a greater willingness in the structures to participate in dialogue rather than to determine that which is right without consultation. In this scenario, the issues over transparency and independence which are raised using the traditional interpretations begin to unravel.

## 5. Conclusions

The probable origin of the word *episkopos* (overseer/bishop) was either an overseer who saw to the distribution of finances and goods in Roman culture (Stewart-Sykes 2014, p. 59) or was derived from the position of the *mebaqqer* (overseer) with a similar function in the Hebrew synagogues (Thiering 1981, p. 66). Similarly, the role of the elder or *presbyterous* (elder/priest), whose advice was sought, having knowledge of the community that was served (Stewart-Sykes 2014, pp. 135–37) in a more egalitarian manner than is present today that can be seen from passages such as Galatians 5:37–38 (Kahl 2014, pp. 192–96), with love's collegiate response being worked out practically in Galatians 5:13–15, as, perhaps, pointed out by Stott (Stott 1992, pp. 158–59). Viewing episcopal church leadership through this lens, the bishop's role of oversight, inherited through the apostolic tradition, appears to have come full circle. From the initial humble beginnings as the authority that distributed the gifts to the needy in the community with the advice of the elders to the present, where control of the dispensation of the diocese's/organisation's funds/ministry is in the bishop's hands, with advice from the diocesan council (or equivalent) fulfilling the role of governance. However, the role has taken on a distinct managerial flavour, as opposed to oversight, which takes up an increasing percentage of the bishop's time (Pickard 2006, p. 23), thus drawing the role away from the ontological prime aspect that has been an acknowledged part of the episcopal role since the days of Ignatius, that of being the president of the Eucharistic assembly. This is not to suggest that the role of the bishop is not at the apex, but rather that the role, perhaps, needs to become more of the original overseer and steersman, in a theological and Eucharistic sense, as opposed to the ultimate manager. In becoming the manager of the enterprise, the bishop is denying the life ($ζωή$) of the body of Christ, as their focus is pulled away from the centrality of the Eucharist and the embodiment of Christ in the community.

Does the interaction between modern (European) philosophy and theology open a new way to perceive ecclesiological governance, at least in the Anglican denomination if not in others? The inheritance of an empiric political governance style has driven the episcopally led church down the route towards sovereign rule, with the episcopacy becoming entrapped by the very authority that sovereignty engenders. This, in turn, disallows the advent of creation and the birth of new life from the chaos of the world. Reliance on the monotheistic centrality without the understanding of what is central to that singularity, that is the understanding borne out of relationship within the Trinity, leads to a managerial present. However, it is perhaps in the interpretation of Romans by Agamben that we may find a way forward for episcopal and Anglican church governance. Agamben speaks about the "state of exception" (Agamben 2005a), which is that state where the law is suspended and, in doing so, the law itself is justified. In turn, this can be suggested to be

similar to the relationship between the Law (the Torah) and the Gospel. Thus, the Gospel imperative becomes the "state of exception" for the Torah, suspending it and yet fulfilling it at the same time (Agamben 2005b, pp. 104–8). In doing so, the potential present in God's word and being is brought back into play so that we do not have to be dependent on or bound by what has been laid down in the law or tradition.

Undoubtedly, the Anglican church and other denominations have been caught in the tension and paradox set out by Paul in Romans. In this context, conforming to the world in obedience to authority (Rom. 13.1-7) rather than being transformed (Rom. 12.2). Thus, our worldly engagement should not be conformance per se, but an engagement with the world from a "different perspective", as suggested by the Gospel imperative of love (Gignac 2013, p. 188). Community without relationship, as Derrida, wisely, is leery about, with its connotations in military fortification, leads to the building of sacrosanct areas of influence. This is, perhaps, where the Agambendian arguments move us towards a weakening of the current understanding of the Anglican church's identity, based on an institution and its legal governance towards a more relational gathering that is based on a governance of love found in the grace of Baptism, guided and not managed by the bishop as a true overseer. Such a governance would be freed to be excessively creative in its expressions of solidarity with the weak, impoverished, and other in the world. In this manner, the centrality of the Eucharist that draws the people of God together becomes, once more, of relevance to the governance of the church, in general, as it responds to the needs of the multitude, rather than to the needs of the law. Practically, this perhaps means that the bishop needs to become more theologically active within the bounds of governance, overseeing and guiding the purposes of God rather than humanity's constructs of a better life. By laying aside the role of the manager and its associated perceived authoritative power, the bishop/clergy become free to express the will of God and its relevance in the community.

The long history of ecclesial governance has been centred around a framing of power that comes from God (unitary, rather than relational) that is conceived in terms of authority with its implementational corollary within the world in terms of action (Deslandes 2018, p. 131). This often leads to a "dictatorship" and autocratic style that has been "prevalent in the Church" (Seoka 1998, p. 101) which can still be seen in some dioceses. The authority focus leads to the attitude of hubris, reflecting a Napoleonic logic of a presumed ability to overcome all obstacles by the person in authority (Kroll et al. 2000, p. 118). Reform means change and it is well known that any change is problematic to an entrenched position unless it is widely embraced by the majority both in power and within the organisation. Traditional governance structures are based on the founding statutes and canons embedded in an understanding of monarchical episcopalism within their core, thus perpetuating the tradition. Reform of such a governance system would have to begin at the core level with a reconceptualization of not only the structure but also the role of power and authority. It is suggested that any such reform would need to start with a deep theological understanding of the role of governance and its purposes for the institution in terms of the church's objective of transforming the world, whilst setting a higher standard than just the conformance that the world requires.

**Funding:** This research received no external funding.

**Data Availability Statement:** No new data were created or analyzed in this study. Data sharing is not applicable to this article.

**Conflicts of Interest:** The author declares no conflict of interest.

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
