# Peer review of "Church Governance—A Philosophical Approach to a Theological Challenge in an Anglican Context"

_religions, doi:10.3390/rel15040427_

Round 1

Reviewer 1 Report

Comments and Suggestions for Authors

This paper has something to offer in that it brings together two ways of thinking about the world, the study of business and theology. I very much value this approach. One of the challenges of this interdisciplinary engagement is in defining terms, providing sufficient background information, and staying free of too much jargon for both disciplines. The author has a good start on some interesting ideas about authority as taken from governance of business organizations and made a start on describing their applicability to church organizations. This is intriguing. As someone with a business background (MBA) and experience in governance, in addition to formal training in theology I very much wanted to dive into the author's work. Very quickly, however, I was bogged down in too many ideas with too little contextual background to understand why these ideas were being discussed and not others. Jargon is overused. Paragraphs are dense and hard to follow. Topic sentences do not assist the reader. I really needed a definition for the word "church" and the specifical social location for which the church and authority were being analyzed. This paper could work, but it needs a much more patient exposition with clearly stated themes and structures. 

Comments on the Quality of English Language

English is used well but there are numerous errors throughout. 

Author Response

Some of the language has been cleared up so as to denote a specific denomination and context. I have tidied up the language where possible but am aware that some may find the phrasing dense. The addressed shortfall in the original will now alleviate the confusion around the contextual background. Jargon is inevitable but I have attempted to alleviate this as much as possible by ensuring that there is a clear definition of specific words and phrases when used for the first time. 

Reviewer 2 Report

Comments and Suggestions for Authors

Here is an outline of your paper as I see it – using your own words whenever possible, and my questions.

1.     “The basis of this governance is founded upon an understanding of a monotheistic all powerful paternal deity, and the Constantinian disposition of power.” Governance in which church? Orthodox? Catholic? Protestant? What century? All churches, at all times?

2.     There are “seven characteristics of good corporate governance”, applicable to all governance. The King Report created by Institute of Directors in South Africa was applicable to all companies listed on the main board of the Johannesburg Stock Exchange. What does this have to do with church governance, Christian and otherwise?

3.     A common progenitor – the status quo of power and authority in the church.

The church’s breakaway from the early egalitarian understanding of governance is found when it joined to empire at the time of Constantine. It is based on the domination of kyriachy (built around domination, oppression and submission.)  Nothing happened in all churches for the last 17 centuries? The Reformation never happened?

4.     An uncommon inheritance – moving into new life. The Anglican / Catholic / Muslim bounded community led Derrida to dislike the term community. The porosity of the modern community is reminiscent of Deleuze and Guattari as they speak of the “rhizome.” A rhizomatic community creates a community that is truly unbound.  Now we see your saviors from the domination of kyriachy: three Parisian intellectuals, Derrida, Deleuze, and Guattari. Can these be seen as representative of “modern (European) philosophy” as it is implied?

5.     Conclusion. The bishop’s role has come full circle. From the initial humble beginnings as the authority that distributed the gifts to the needy to the present where control of the dispensation of organization’s funds is in the bishop’s hands. Nothing happened in the last 17 centuries? Calvinist churches have no bishops.

     The word “Anglican” appears four times: in the abstract, as a key word, in “Anglican/Catholic/Muslim,” and in “the Anglican polity (into which polity I fall).”  Who falls into the Anglican polity? All the subjects of his Majesty the King as head of the Anglican Church? What is your relationship with and knowledge of the Anglican Church of England, besides falling into its polity?

Author Response

The response to the reviewers specific questions and comments are in the attached document.

Reviewer 3 Report

Comments and Suggestions for Authors

This essay argues that the governance of churches should not be exclusively "managerial" (following the practices of any business or social organization), but should also allow for non-hierarchical change and exploration, opening churches to less predictable influences of spiritual insights from its leaders and members. In short, the church is not a business, but a community of beings in ongoing relation to the divine. It is certainly a point well worth considering by the readers of this journal. It might be objected that in addition to being open to inspiration, any church nonetheless needs to follow some set of principles of organization if it is to exist at all, but presumably the author(s) would maintain that this matter is not an either/or, and they are recalling church leaders to their original purpose, while also maintaining an organization. It is a thesis well worth promoting.

Comments on the Quality of English Language

No serious language issues are detected, but some proofreading might fix minor lapses in punctuation and sentence structure.

Author Response

I thank the reviewer for their response and have edited minor English throough out as much as possible

Round 2

Reviewer 1 Report

Comments and Suggestions for Authors

My primary concern with the paper is its sweeping claims about the origin of church governance. References are made to historical events and scripture, but the paper's claims are neither grounded in historical research or scriptural exegesis.  That is, there is no evidence in the paper that what the author claims about the current church structure or development actually occurred in history or that it stems from scripture.   For this reason, the current version of the paper likely does not contain truth claims about how the church is or was. However, the lack of such truth claims does not keep the paper from saying something interesting and thought provoking. There is a very brief acknowledgement in the opening that the paper is providing a lens for reflection. I take this to mean that the paper is provoking reflection, though its correspondence to actual conditions may be true only in the most abstract way. 

Given my comments above, section 4 is very problematic. First, its connection to the definition of governance described earlier is difficult to discern. But, more importantly, the section makes claims about how things actually function in the Anglican church without any justification (historical or phenomenological) that what the author states is true. For me, this section thus becomes an elaborate word game or argument by association. 

I do think there is something here to say about governance. Something important. I am not convinced that connecting it to European philosophy is the best way to bring it forward.  

Author Response

see attached document
